# A study of spatial distribution and dynamic change in monthly FVC of urban parks

**Yichuan Zhang**[1,2]*, **Yanan Ge**[1], **Lifang Qiao**[1]

**1** School of Horticulture and Landscape Architecture, Henan Institute of Science and Technology, Xinxiang, China, **2** Henan Province Engineering Center of Horticultural Plant Resource Utilization and Germplasm Enhancement, Xinxiang, China

* zhangyichuan@hist.edu.cn

**Data Availability Statement:** Data relevant to this study are available from GitHub at https://github.com/zhangyichuan2002/PONE-D-24-17066.

**Funding:** This research was funded by the following projects: Key Science and Technology Research and Development Program of Henan

## Abstract

The study on the spatial distribution and dynamic change in monthly Fractional Vegetation Cover (FVC) of parks provides a scientific basis for vegetation management and optimization in urban parks. This research focuses on two comprehensive parks located in Xinxiang, China—People's Park and Harmony Park, using multi-spectral Unmanned Aerial Vehicle (UAV) images as the data source and considering monthly periods. Monthly FVC data was obtained using the method of Dimidiate Pixel Model based on the Normalized Difference Vegetation Index (NDVI). The dynamic changes of monthly FVC at regional scale were described through the dynamic changes in the monthly FVC mean and in the FVC areas at various scales, and the dynamic changes in the monthly FVC were analyzed using the coefficient of variation and curve change trends. Furthermore, the dynamic changes in FVC areas at various scales in the parks were analyzed using standard deviation and curve change trends. Subsequently, the differential method was used to analyze the monthly FVC dynamic changes at pixel scale. The results indicate: (1) In terms of the spatial distribution characteristics in monthly FVC of urban parks, both parks exhibit the highest ratio of bare area in January and February. The proportions of FVC for People's Park are 59.17% and 64.46%, while for Harmony Park they are 69.10% and 51.92%, showing the most distinct spatial distribution characteristics. The high and very high coverage areas in each month are mainly distributed on the outskirts of the park, while the medium, medium-low, and low coverage areas are mainly located in the central and middle parts of the park. The overall FVC of the park shows a trend of high coverage on the periphery and low coverage in the center. (2) In the spatial-temporal dynamic change in FVC at regional scale, the average monthly FVC changes exhibit an overall "∩" -shaped pattern. The peak and minimum FVC values for different parks occur at different times. The peak FVC for People's Park appears in August, while for Harmony Park it appears in June, with corresponding FVC values of 0.46 and 0.50, respectively. The minimum FVC for People's Park occurs in February, and for Harmony Park it occurs in January, with FVC values of 0.17 and 0.15, respectively. Among the dynamic change in FVC areas at various scales, the areas of bare and highest-coverage exhibit the greatest fluctuations, with the ascending and descending changes and rates of bare and highest-coverage areas generally showing opposite trends. (3) In terms of the spatial-temporal dynamic changes in FVC at pixel scale in urban parks, overall, FVC

Province, China (232102320022), and Key Science and Technology Research and Development Program of Henan Province, China (232102320071). The funders had no role in study design, data collection and analysis, decision to publish, or preparation of the manuscript.

**Competing interests:** The authors have declared that no competing interests exist.

shows moderate improvement from February-August, and moderate degradation from January-February and from August-December. The degradation and improvement are primarily slight. The most significant improvement in monthly FVC occurs in March-April, with a predominant type of significant improvement in FVC changes. People's Park and Harmony Park show the most significant degradation in FVC during September-October and October-November, respectively, with a predominant type of significant degradation in FVC changes. During the periods of most significant improvement and degradation in monthly FVC, the spatial distribution of significant improvement and degradation areas primarily occurs in the periphery and middle parts of the parks. FVC in urban parks decreases from January to February and from August to December, while it increases from February to August, with relatively good conditions from June to August. Vegetation optimization should consider: balancing recreational and ecological functions overall, controlling the proportion of bare land, and enhancing the canopy structure of vegetation in low coverage areas or the coverage of hard surfaces; locally increasing the proportion of evergreen plants and moderately increasing planting density. In addition, parks should strengthen management to reduce the impact of flooding and maintain the health of vegetation.

## Introduction

Urban green spaces are essential ecological infrastructure within cities, playing a significant role in maintaining ecological balance, improving environmental quality, and promoting resident health [1, 2]. Vegetation serves as the primary carrier of ecological benefits in urban green spaces, with the NDVI being one of the most commonly used vegetation indices for assessment [3]. Studies indicate that the proportion of vegetation, water, and settlement in urban land use directly influences land surface temperature (LST) [4]. Additionally, there is a certain correlation between NDVI and LST, and they exhibit different ecological roles in different seasons [5, 6]. NDVI demonstrates saturation, meaning that it is influenced by soil and canopy background, and does not continue to increase with plant growth once vegetation density reaches a certain grade [7]. FVC is an important ecological indicator in the assessment of urban green spaces [8]. FVC refers to the percentage of the area occupied by vegetation in the vertical projection area per unit area, directly reflecting the vegetation status of green spaces [9]. Research indicates that FVC in urban green spaces significantly influences ecosystem services such as carbon absorption [10], biodiversity conservation [11], and soil moisture regulation [12]. Increasing FVC can enhance the urban ecological environment, mitigate the urban heat island effect [13], improve air quality [14], and promote water resource circulation, providing crucial support for sustainable urban development. FVC in urban green spaces also has a significant impact on resident health, as higher FVC in green spaces contributes to stress reduction, anxiety alleviation, and improved psychological well-being [15, 16]. Additionally, urban green spaces with higher FVC contribute to improving residents' quality of life [17] and promoting social cohesion [18]. In recent years, scholars have begun to focus on the spatial-temporal dynamic change in FVC of urban green space [19], exploring the impact of factors such as urban expansion and land use change on FVC [20], thereby providing important reference points for urban green space planning and management.

In recent years, with the development of remote sensing technology and in-depth research on urban green space management, significant progress has been made in FVC-related studies.

Monitoring techniques for FVC have not only continued to innovate in remote sensing technology [21, 22], but have also explored new methods based on ground observation and UAV technology, improving the spatial-temporal resolution and accuracy of FVC monitoring [23]. Compared to traditional aerial remote sensing, UAVs have the advantages of flexibility and efficiency, enabling flexible flight planning and control as needed to achieve rapid and precise monitoring of urban green spaces, significantly enhancing research efficiency [24]. UAVs equipped with high-resolution cameras can obtain high-quality image data of urban green spaces, providing clear ground information and capturing vegetation structure and spatial distribution at a microscopic scale [25]. When UAVs are equipped with multi-spectral sensors, they can obtain multi-band image data of urban green spaces, providing spectral information of vegetation to aid in identifying different types of FVC and conducting quantitative analysis, thereby improving the precision and accuracy of FVC measurement [9]. Due to the real-time and dynamic nature of UAVs, they can achieve real-time monitoring and dynamic change analysis of FVC in urban green spaces, facilitating timely understanding of changes in urban green spaces and providing timely support for urban green space management and emergency response.

Urban parks serve as vital venues for leisure and recreational activities for city residents [26]. Vegetation, as a primary component of natural park landscapes, significantly impacts the physical and mental well-being of individuals [27]. High FVC can increase oxygen release, absorb harmful gases and particulate matter in the air, thereby improving air quality within and around the park, providing residents with a fresh air environment [28]. Parks with high FVC exhibit rich natural charm and green landscapes, adding unique ecological allure and cultural ambiance to the city, enhancing its scenic quality and attractiveness [29]. Dense vegetation cover also provides abundant habitats and food resources, attracting a variety of wildlife and promoting biodiversity [30]. Therefore, effectively increasing FVC of urban parks can create more aesthetically pleasing and ecologically sound leisure and recreational spaces, offering residents a comfortable and pleasant living environment [31].

Previous research has often been conducted on an annual basis and at a relatively large spatial scale. Parks, as crucial ecological infrastructure scattered throughout cities, play important roles in leisure, recreation, and social activities [32]. Parks are typically of small size, diverse in function and style, and rich in variety. However, from a physical structural perspective, parks consist of landscape elements such as trees, shrubs, grasslands, water bodies, hard paving, and roads, albeit in different combinations and proportions. Among these elements, vegetation is the primary component through which parks deliver ecological benefits, and its combination with other elements creates high heterogeneity. Therefore, low-resolution images fail to meet the precision requirements for analysis.

Months represent typical units for the phenological periods of plants within a year. Monitoring vegetation phenology over short time periods aids in detecting rapid changes in vegetation [33]. The FVC in park naturally changes with the succession of months. This study aims to address the following issues using high-precision multi-spectral UAV images: How does the FVC differ in different spatial locations each month, and to what extent? How does the FVC change within the same spatial location between months, and to what extent? By analyzing the spatial distribution and dynamic changes in park FVC, how to provide a basis for optimizing park vegetation from a planning and design perspective.

## Materials and methods

### Research area and data sources

**Research area.**   Xinxiang is located in the northern part of Henan Province, China, with geographic coordinates ranging from 113°23′ to 115°01′ east longitude and 34°53′ to 35°50′

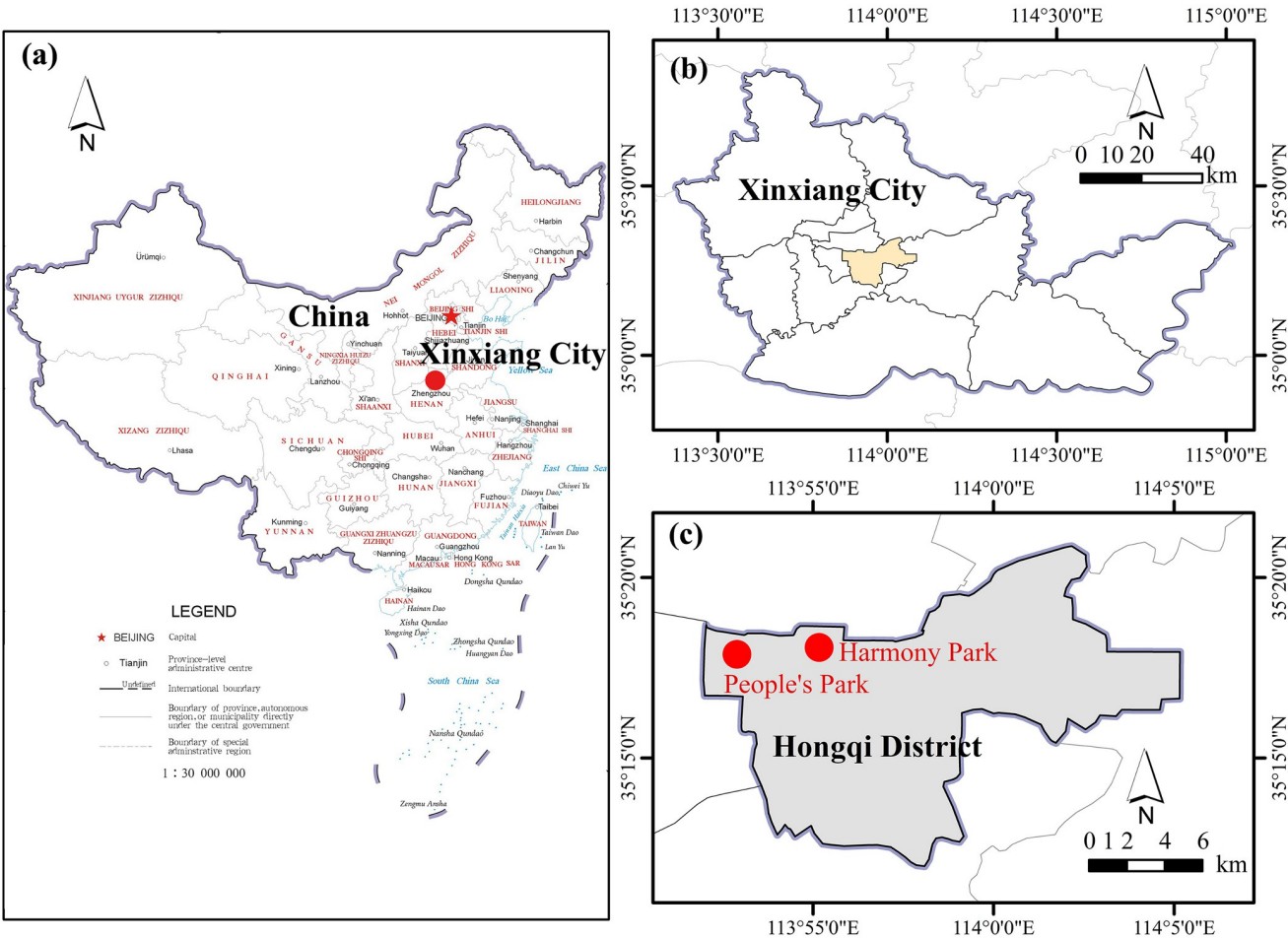

**Fig 1.** (a) Geographic Location of China (Map sourced from the Ministry of Natural Resources of China, Carto-graphic License: GS (2019)1659); (b) Geographic Location of Xinxiang City; (c) Geographic Location of Hongqi District in Xinxiang City and the Two Parks.

north latitude, covering a total area of 8,249 square kilometers. The city accommodates several dozen parks, among which People's Park and Harmony Park are the most representative. People's Park, established in 1956, is the earliest park in Xinxiang and the largest comprehensive park integrating garden landscaping, water entertainment, cultural fitness, and leisure activities. Harmony Park, built in 2005, is the city's first modern park, primarily emphasizing ecological landscapes and leisure and recreational functions (Fig 1). These two parks both provide rich activity spaces for the public, including water bodies, plazas, and roads. They are characterized by lush vegetation, complex plant community structures, and a wide variety of plant species. These plants are combined with activity spaces, forming highly heterogeneous vegetative landscapes.

**Data sources and processing.** A DJI Phantom 4 Multispectral UAV, featuring an integrated multispectral imaging system, was used to capture monthly time-series images of the parks from January to December 2021 (Table 1). The imagery data underwent single-band image stitching to obtain overall images of five bands: red, blue, green, near-infrared, and red edge. Data analysis and mapping were carried out using ENVI 5.3 and ARCGIS 10.8 software.

Table 1. Image data acquisition parameters.

| Parameter | Name | |
|---|---|---|
| | People's Park | Harmony Park |
| Coordinate System | WGS84 /UTMzone 49N | WGS84 /UTMzone 49N |
| Flight Altitude | 150m | 150m |
| Along-Track Overlap | 80% | 80% |
| Across-Track Overlap | 80% | 80% |
| Main Track Angle | 0˚ | 0˚ |
| Weather Conditions | Windless and cloudless | Windless and cloudless |
| Camera Model Name(s) | FC6360_5.7_1600x1300 | FC6360_5.7_1600x1300 |
| Average Ground Sampling Distance (GSD) | 8.10 cm/3.19 in | 8.03 cm/3.16 in |
| Dataset | 984 images | 230 images |
| Kappa | Mean(0.006), Sigma(0.001) | Mean(0.005), Sigma(0.001) |

## Research methods

The research framework can be seen in the diagram (Fig 2).

**NDVI data sources.** NDVI was derived by calculating the difference between the near-infrared and red bands with the following formula:

$$NDVI = (NIR - R)/(NIR + R) \tag{1}$$

Where NIR denotes the reflectivity of the near-infrared band, while R indicates the reflectivity of the red band.

Since the NDVI value of the water body is less than 0, in order to avoid disturbing the value, this study counts it into the bare area after zeroing it through ENVI 5.3 software.

**FVC calculation and classification.** The Dimidiate Pixel Model is a simple and direct method for estimating Fraction of FVC based on NDVI [34]. The model assumes that the land surface of a pixel consists of both green vegetation and bare soil, with FVC representing the proportion of vegetation within the pixel. FVC is calculated using the formula [35]:

$$FVC = (NDVI - NDVI_{\text{soil}})/\left(NDVI_{veg} - NDVI_{\text{soil}}\right) \tag{2}$$

Where $NDVI_{\text{soil}}$ denotes the NDVI value for a pixel representing pure bare area under ideal conditions, while $NDVI_{veg}$ corresponds to the NDVI value for a pixel representing area completely covered by vegetation under ideal conditions. Pure bare ground cover refers to areas completely covered by soil without any vegetation, while pure vegetation cover refers to areas devoid of any soil coverage [36]. Theoretically, these values approach 0 and 1, respectively, in an ideal scenario. However, due to various influencing factors, the actual minimum and maximum NDVI values may not align with $NDVI_{\text{soil}}$ and $NDVI_{veg}$. Drawing on prior research [37], this study adopted the NDVI values of 5% and 95% for pure bare area and area completely covered by vegetation, respectively.

In accordance with the SL190-2007 Standards for Classification and Gradation of Soil Erosion from the Ministry of Water Resources of the People's Republic of China, FVC grades were categorized as: bare area (FVC ≤ 10%), lowest coverage area (10% < FVC ≤ 30%), lower coverage area (30% < FVC ≤ 45%), middle coverage area (45% < FVC ≤ 60%), higher coverage area (60% < FVC ≤ 75%), highest coverage area (FVC > 75%).

**Monthly FVC spatial distribution characteristics.** The FVC values for each month in the parks were obtained based on 12 months of multi-spectral UAV data. To characterize

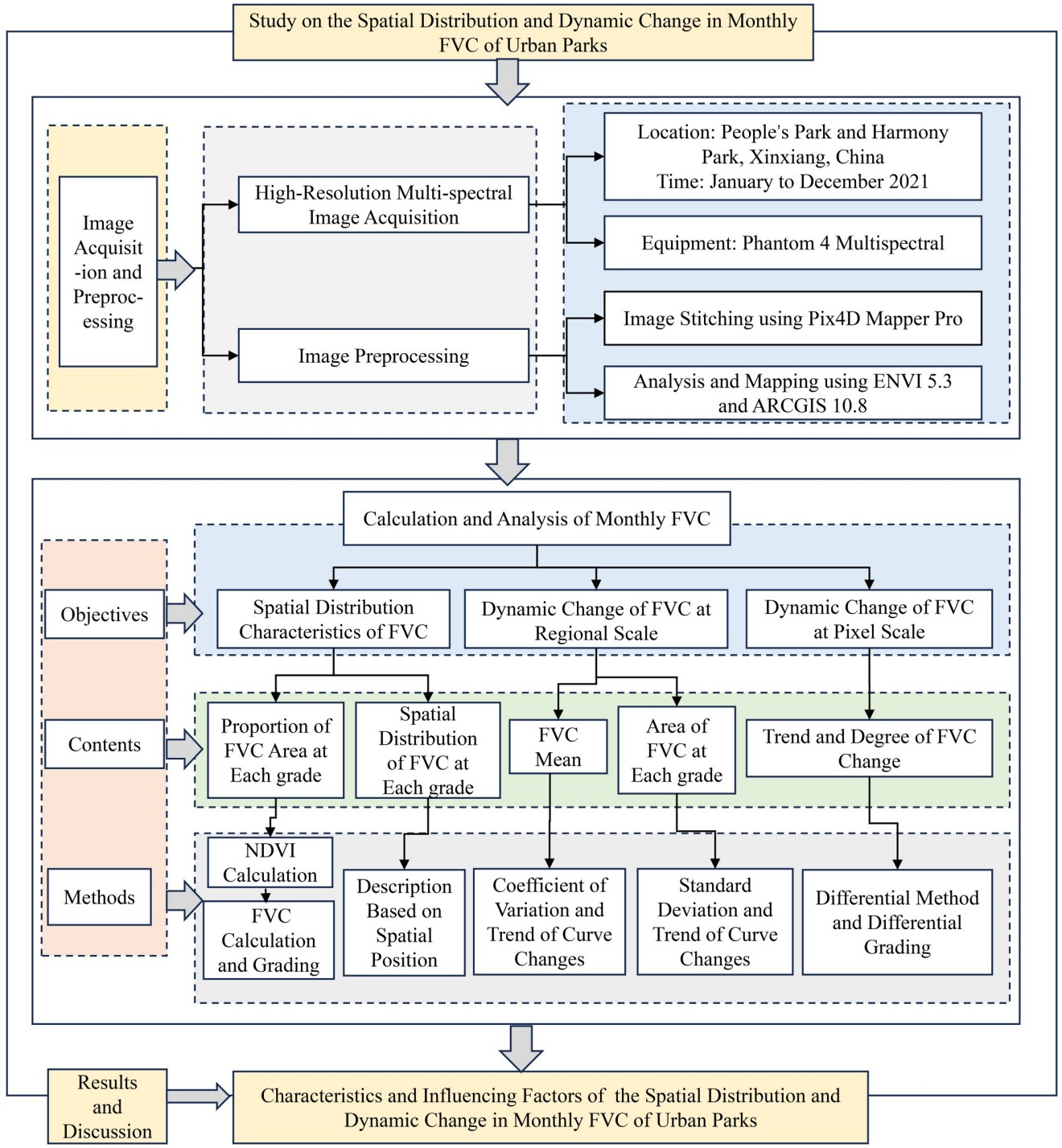

**Fig 2. Research framework diagram.**

spatial distribution, the parks' boundaries were incrementally contracted to create three concentric rings, delineating the spatial location into the peripheral, middle, and central sections.

**Spatial-temporal dynamic change in monthly FVC at regional scale.** The spatial-temporal dynamic changes in FVC at regional scale include dynamic changes in FVC mean values

and dynamic changes in FVC area at various grades. The coefficient of variation can be used to compare the dispersion of data from different groups. This paper uses the coefficient of variation to compare the fluctuation of monthly FVC mean values in the two parks. Additionally, combined with the curve of monthly FVC values, the appearance times of the maximum and minimum FVC values are analyzed. The standard deviation is widely used in cases where precise measurement of data dispersion is needed. In this paper, the standard deviation is used to compare the dispersion of FVC area at various grades for each park, expressing the fluctuation of FVC area at different grades. Combined with the curve of monthly FVC area at various grades, the changes in FVC at various grades between months are analyzed.

The categorization of FVC stability in this paper is divided into three grades: weak variation (C < 0.1), moderate variation (0.1 ≤ C < 1), and strong variation (C > 1). The formula is as follows [38]:

$$C = \sigma/\bar{C} \tag{3}$$

Where σ represents the standard deviation, and $\bar{C}$ represents the mean value.

The standard deviation σ reflects the precise dispersion of FVC area at various grades. The formula is as follows:

$$\sigma = \sqrt{\frac{\sum_{i=1}^{n}\left(C_i - \bar{C}\right)^2}{n}} \tag{4}$$

Where $C_i$ represents the area of a certain FVC grade for a month, $\bar{C}$ is the average of the areas of that FVC grade across 12 months, and n represents the number of months.

**Spatial-temporal dynamic change in monthly FVC at pixel scale.** The differential method was employed to examine the spatial-temporal dynamic change in monthly FVC at pixel scale. To graphically depict the pixel-scale enhancements and deteriorations of monthly FVC, Based on the calculated differences, the ENVI 5.3 software was used for classification. Drawing on prior studies and in-situ vegetation assessments, vegetation changes were categorized into different grades. The difference was determined using the layer overlay and band math functions in ENVI 5.3 software, as per the following formula [39]:

$$\Delta FVC = FVC_i - FVC_{i-1} \tag{5}$$

Where ΔFVC is the FVC difference index, ΔFVC > 0 means FVC increases while ΔFVC < 0 means FVC decreases; $FVC_i$ and $FVC_{i-1}$ are the pixel values of the FVC images of the two adjacent months respectively. Subsequently, the dynamic changes in FVC were divided into 5 grades: significant degradation (less than -30%), slight degradation (-30% to -5%), basic stability (-5% to 5%), slight improvement (5% to 30%), and significant degradation (greater than 30%).

## Results

### Monthly FVC spatial distribution characteristics

**Analysis of the ratio area of monthly FVC at each grade.** The proportions of FVC area for each grade across months in Xinxing Park are illustrated in Fig 3. Overall, the bare area had the largest ratio in People's Park for the months of January to December. Harmony Park, on the other hand, showed the highest coverage area ratios in May, June, and August, with the remaining nine months having the largest ratio of bare area.

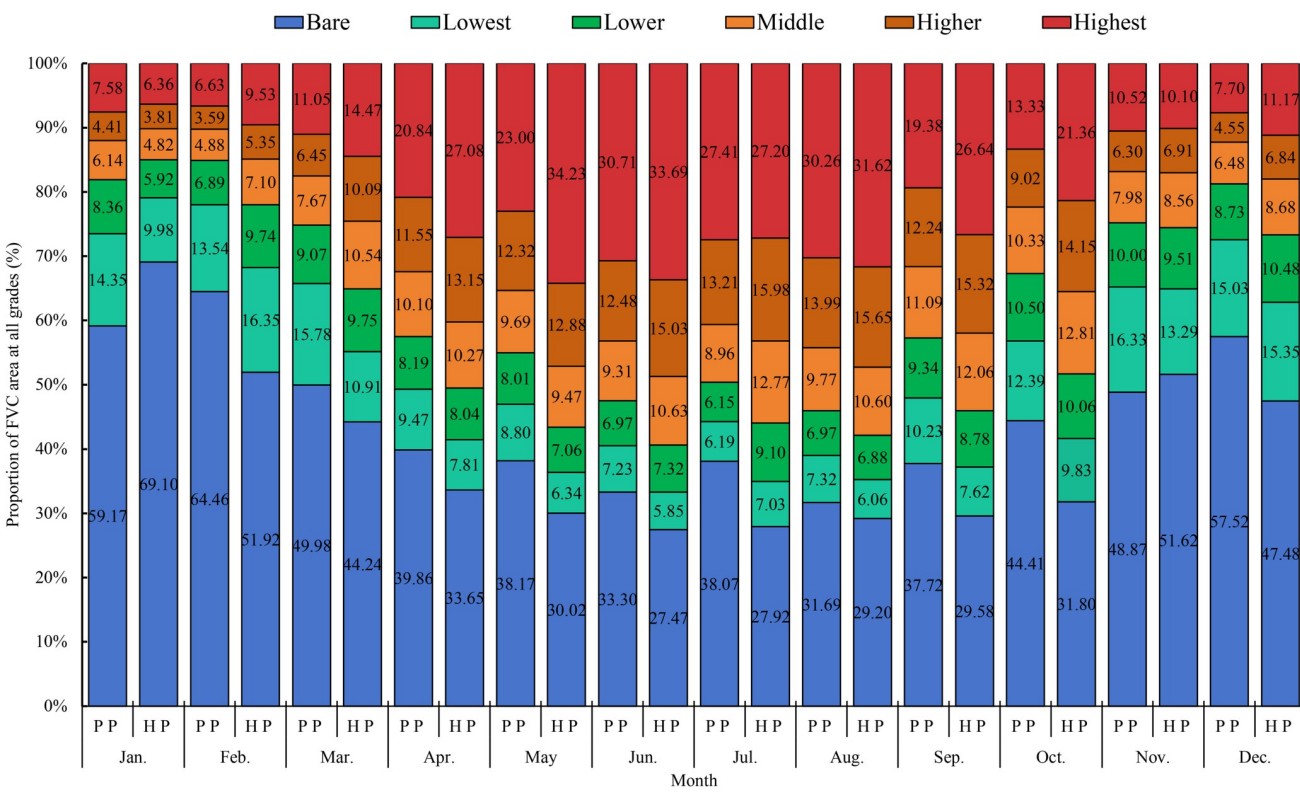

**Fig 3. Area ratio map of monthly FVC at each grade (%).** (Note: People's Park is abbreviated as P P, and Harmony Park is abbreviated as H P).

In People's Park, for January and February, the ratio areas of FVC at each grade followed the same coverage order, namely: bare, lowest coverage, lower coverage, highest coverage, middle coverage, and higher coverage. For March, the order was bare, lowest coverage, highest coverage, lower coverage, middle coverage, and higher coverage. From April to September, the ratio areas of FVC at each grade maintained a consistent order for six consecutive months, namely: bare, highest coverage, higher coverage, middle coverage, lowest coverage, and lower coverage. For October, the order was bare, highest coverage, lowest coverage, lower coverage, middle coverage, and higher coverage. In November, the order was bare, lowest coverage, highest coverage, lower coverage, middle coverage, and higher coverage. For December, the order was bare, lowest coverage, lower coverage, highest coverage, middle coverage, and higher coverage. Notably, in January, February, March, November, and December, the ratios of bare and lowest-coverage areas were relatively large, while for the continuous seven months from April to October, the ratios of bare and highest-coverage areas were relatively large.

In Harmony Park, the ratio areas of FVC at each grade followed the same coverage order for January, November, and December, which was: bare, lowest coverage, highest coverage, lower coverage, middle coverage, and higher coverage. For February, the order was bare, lowest coverage, lower coverage, highest coverage, middle coverage, and higher coverage. In March, the order was bare, highest coverage, lowest coverage, middle coverage, higher coverage, and lower coverage. For April, July, September, and October, the order was: bare, highest coverage, higher coverage, middle coverage, lower coverage, and lowest coverage. For May, June, and August, the order was highest coverage, bare, higher coverage, middle coverage,

lower coverage, and lowest coverage. Notably, in January, February, November, and December, the ratios of bare and lowest-coverage areas were relatively large. From March to October, the ratios of bare and highest-coverage areas were relatively large for eight consecutive months. Among these, in May, June, and August, the highest-coverage area had the largest ratio, while in April, July, September, and October, the bare area had the largest ratio. Comparatively, within a single year, the ratio areas of FVC for the twelve months fluctuated significantly in Harmony Park, while the trends in the ratio areas of FVC in People's Park were more stable. This indicated that different city parks exhibit varying trends in the ratio areas of different FVC grades within the same time period.

**Monthly FVC spatial distribution characteristics at each grade.** The ratios of FVC at each grade varied across months in urban parks, resulting in different spatial distribution characteristics of FVC (Fig 4). Generally, in the period from January to March, the areas with medium-high and high FVC coverage in both parks are mainly located on the periphery, gradually spreading towards the central areas as the months progress. During this period, the "bare ground-medium coverage area" is mainly distributed in the central and middle parts of the parks. From April to June, the areas with medium-high and high FVC coverage are mainly distributed in the periphery and middle regions of the parks, while the "bare ground-medium coverage area" is primarily concentrated in the central and middle parts of the parks. During this period, the areas with me-dium-high and high coverage change with the months, gradually expanding towards the park's center. By June, these areas have spread across the entire park. From July to September, the areas with medium-high and high FVC coverage are widely distributed throughout the parks, while the "bare ground-medium coverage area" has a smaller range, scattered across the central and middle parts of the parks. Specific analysis based on Fig 3 reveals that from July to August, the areas with medium-high and high coverage show an expanding trend, spreading from the periphery towards the center of the parks. Conversely, from August to September, the areas with medium-high and high coverage show a decreasing trend, while the "bare ground-medium coverage area" exhibits an expanding trend, spreading from the park's center towards the middle and outer parts. From October to December, the areas with medium-high and high FVC coverage are mainly distributed in the periphery of the parks. As the months progress, the areas with medium-high and high coverage decrease in range. Simultaneously, the "bare ground-medium coverage area" expands towards the periphery of the parks, and by November and December, it has essentially covered the entire parks. Overall, areas with higher coverage and highest coverage were mainly distributed at the periphery of the parks, with continuous and concentrated distribution, and a small amount located at the park center. Meanwhile, areas with middle coverage, lower coverage, and lowest coverage were primarily situated at the park center and middle parts, appearing more scattered. Bare area was mainly found in the water bodies and hard surfaces. The FVC spatial distribution across the 12 months in urban parks generally exhibited a pattern of higher values at the periphery and lower values at the center.

## Spatial-temporal dynamic change in monthly FVC at regional scale

**Dynamic change in monthly FVC mean.** Fig 5 illustrated the dynamic changes in the monthly FVC means of two parks. The monthly distribution of FVC approximates a normal curve, showing an overall trend of increase followed by decrease. In the monthly FVC means, except for January, the FVC means for each month in People's Park are lower than those in Harmony Park. The coefficients of variation for the monthly FVC means in People's Park and Harmony Park are 0.3248 and 0.3197, respectively, indicating considerable fluctuations in the FVC means for both parks.

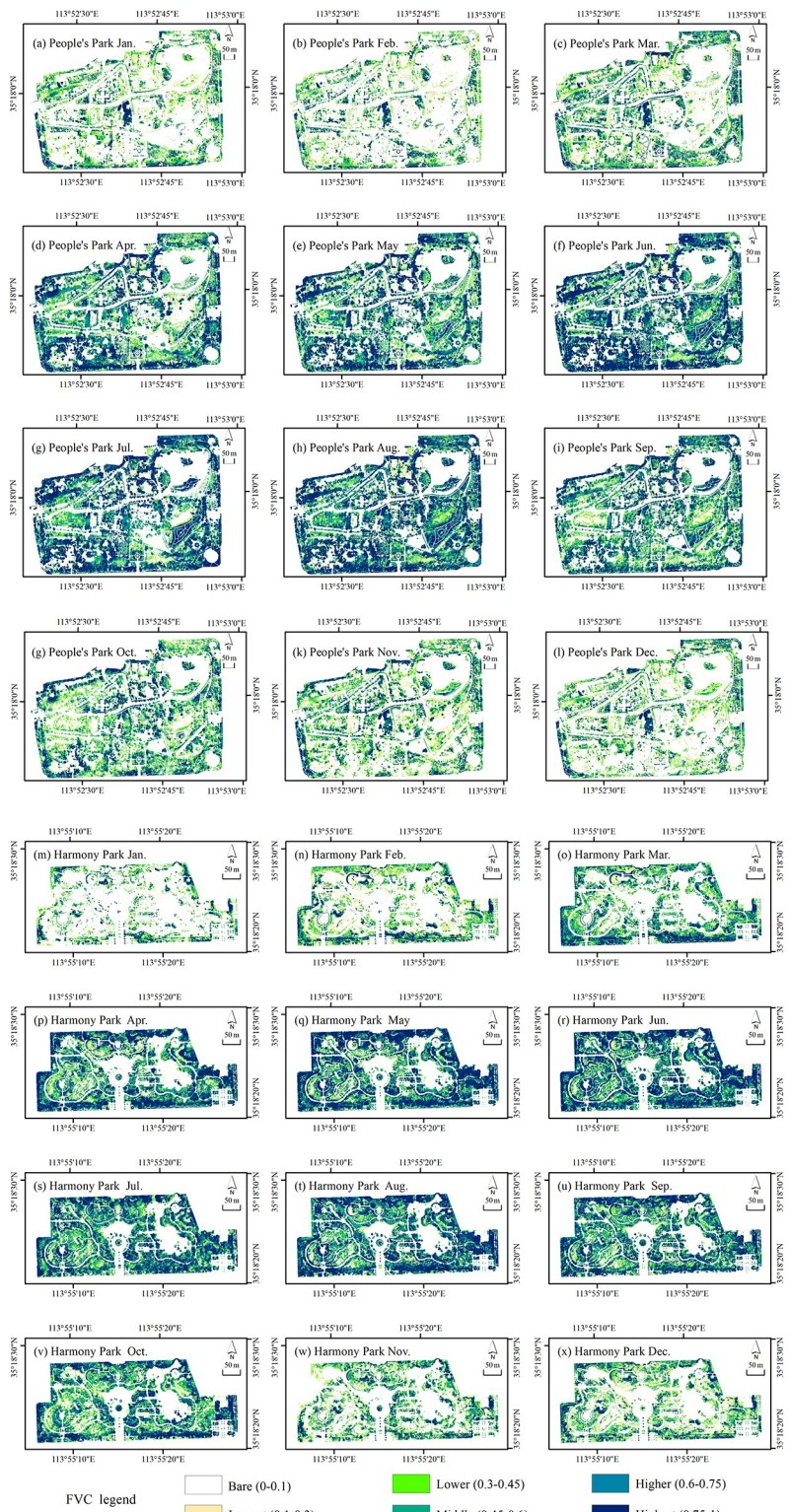

**Fig 4. Monthly FVC spatial distribution map: (a-l) People's Park 1–12 months; (m-x) Harmony Park 1–12 months.**

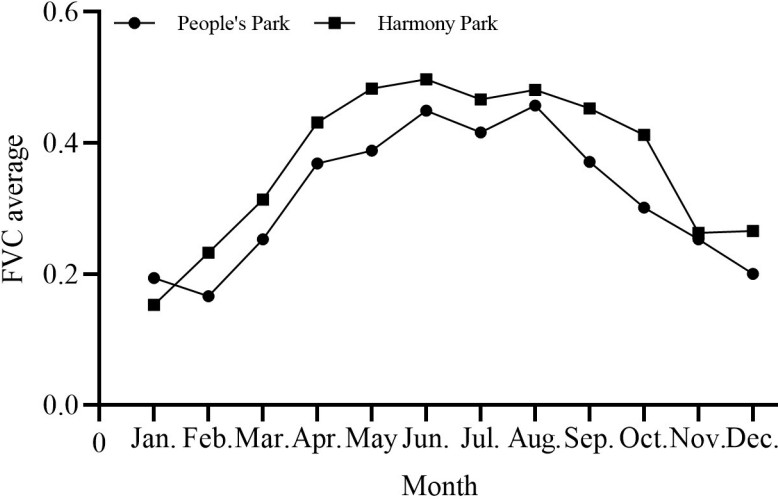

**Fig 5. Tendency chart of monthly FVC mean.**

Over the course of a year, with the exception of January, the FVC means for each month in People's Park were lower than those in Harmony Park. The rate of change in monthly FVC mean is 0.004821/a for People's Park and 0.007941/a for Harmony Park, indicating a higher rate for Harmony Park. Overall, the monthly FVC mean for both parks showed an increasing trend from January to June, reaching a peak from June to August, and then a decreasing trend from August to December. The ranking of monthly FVC mean for People's Park was as follows: August > June > July > May > September > April > October > November > March > December > January > February, with the highest FVC average in August at 0.46 and the lowest in December at 0.17. For Harmony Park, the ranking was: June > May > August > July > September > April > October > March > December > November > February > January, with the highest FVC average in June at 0.50 and the lowest in January at 0.15. The differences in the rate of change and average rankings of monthly FVC between the two parks indicated distinct vegetation growth characteristics in different spatial locations over the same time period.

**Dynamic change in monthly FVC area at each grade.**  The dynamic change characteristics of the FVC area at each grade (Fig 6). Overall, in the dynamic change of FVC area across different grades for both parks, the standard deviations of the bare ground and high coverage areas for each month in People's Park are 5.123 and 4.325, respectively, which are higher than the standard deviations for other grades. In Harmony Park, the standard deviations of the bare ground and high coverage areas for each month are 1.445 and 1.120, respectively, also higher than the standard deviations for other grades. Therefore, in the monthly dynamic changes of FVC area for both parks, the areas with bare ground and high coverage exhibit the greatest fluctuations, while other grades remain relatively stable. In the monthly changes in FVC area at each grade, the area fluctuations of bare and highest-coverage areas generally exhibited opposite trends.

The analysis revealed the following: In People's Park, the bare area increased in the first two months, then steadily decreased from February to June, fluctuated irregularly from June to August, and continuously increased from August to December. Specifically, the trends of increase and decrease in the area of bare were stable between February to June and August to December. In Harmony Park, the bare area consistently decreased from January to June,

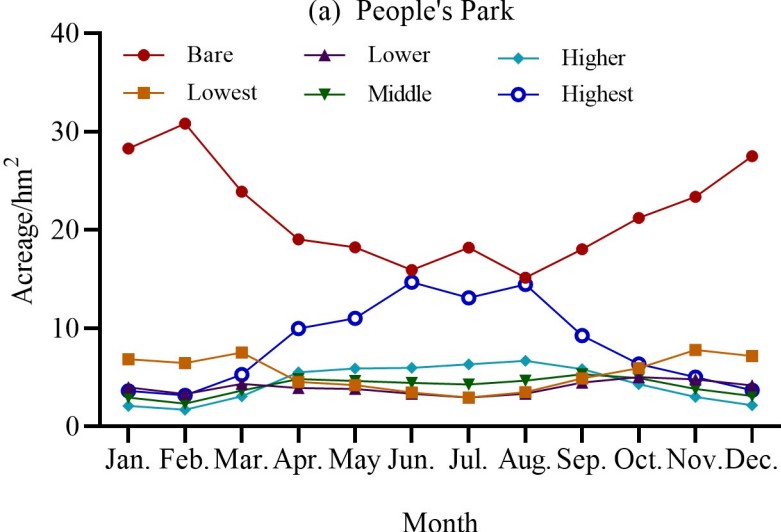

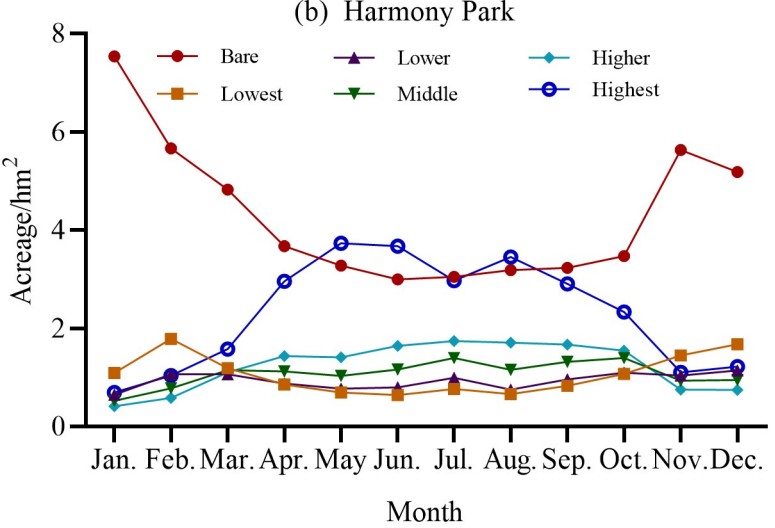

**Fig 6. Dynamic tendency chart of monthly FVC area at each grade: (a) People's Park; (b) Harmony Park.**

steadily increased from June to November, and showed a decreasing trend from November to December.

Regarding the changes in the highest-coverage area, in People's Park, the variations in the highest-coverage area exhibited a completely opposite trend to those of the bare area. The highest-coverage area decreased from January to February, steadily increased from February to June, fluctuated significantly from June to August (decreasing from June to July and increasing from July to August), and continuously decreased from August to December. The trends of increase and decrease in the highest-coverage area were stable between February to June and August to December. In Harmony Park, the highest-coverage area showed an increasing trend from January to May, a decreasing trend from May to July, an increasing trend from July to August, a decreasing trend from August to November, and an increasing trend from November to December. The highest-coverage area exhibited significant

fluctuations, with relatively stable trends between January to May and August to November. A comprehensive analysis of the relationship between the area of bare and highest-coverage areas in the parks revealed that in People's Park, the area of bare was closest to the area of highest-coverage from June to August, although the area of bare consistently exceeded the highest-coverage area. In Harmony Park, the area of bare was closest to the area of highest-coverage from May to September, and in May, June, and August, the highest-coverage area exceeded the bare area.

## Spatial-temporal dynamic change of monthly FVC at pixel scale

**Analysis of monthly FVC change types.** Area ratio map of different monthly FVC change types is shown in Fig 7. Overall, for both parks, the main FVC change type across the 12 months was basically stability. Apart from this, the neighboring months in both parks showed the following trends of improvement and degradation: Improvement was predominant in the periods of February-March, March-April, April-May, May-June, and July-August, while degradation was predominant in the periods of January-February, June-July, August-September, September-October, and October-November. For People's Park, besides the basic stability area, the top three ratios of the improvement area were as follows: March-April > May-June > February-March, with improvement area ratios of 40.36%, 39.52%, and 39.46% respectively. The top three ratios of the degradation area were: August-September < September-October < November-December, with degradation area ratios of 45.11%, 38.71%, and 33.86% respectively. For Harmony Park, apart from the basic stability area, the top three periods with the largest ratios of improvement were: March-

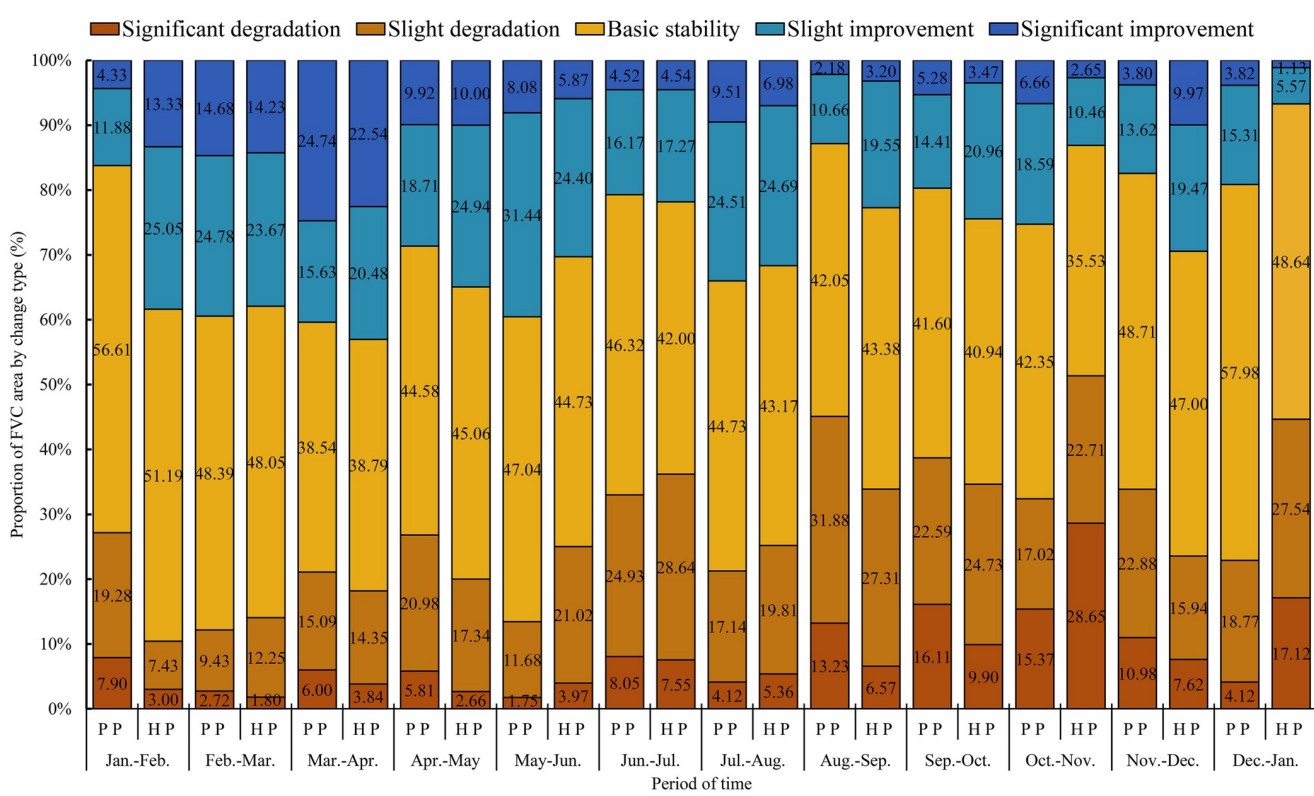

**Fig 7. Area ratio map of different monthly FVC change types (%).** (Note: People's Park is abbreviated as P P, and Harmony Park is abbreviated as H P).

April > January-February > February-March, with improvement area ratios of 43.02%, 38.38%, and 37.91% respectively. The top three periods with the largest ratios of degradation were: October-November < December-January < September-October, with degradation area ratios of 51.36%, 44.66%, and 34.63% respectively.

The specific change types corresponding to the adjacent months' FVC changes were as follows: For all 12 months, the basic stability area was predominant. Besides this, in People's Park, the periods of February-March, March-April, April-May, May-June, and July-August, showed a trend of improvement, with March-April having a significantly larger ratio of the significant improvement area, and April-May having a relatively larger ratio of slight degradation area, while the other three periods had a relatively larger ratio of slight improvement area. For Harmony Park, apart from the basic stability area, the FVC was predominantly improving in January-February, February-March, March-April, April-May, May-June, July-August, and November-December. March-April showed a significant improvement, and the other six periods showed a slight improvement. FVC is predominantly degrading in January-February, June-July, August-September, September-October, and October-November, with October-November showing a significant ratio of significant degradation area, while the others showed a relatively larger ratio of slight degradation area. In general, apart from the basic stability area, the predominant FVC changes in both parks were slight improvement and slight degradation, with significant improvement and significant degradation areas occupying relatively smaller ratios. This indicated a relatively stable monthly FVC change at pixel scale.

**Spatial distribution characteristics of monthly FVC main change types.** The dynamic changes in the monthly FVC at pixel scale are depicted in Fig 8. Overall, from February to August, both parks show improvement as the main trend. During this period, significant improvements occur from February to June, covering most areas of the parks. There are also some areas where the trend of improvement or degradation differs from the overall monthly trend, mainly in areas with evergreen vegetation. The areas of improvement from June to August are mainly scattered. In People's Park, the areas of improvement from June to August are concentrated on the slopes in the middle of the park, while in Harmony Park, the areas of improvement are scattered around the periphery of the park. From August to December, the vegetation in both parks generally shows a trend of degradation, with diverse spatial distribution character-istics. However, overall, degradation occurs in multiple areas throughout the parks simultaneously. Similarly, there are regions from August to December where the trend contradicts the overall trend of improvement or degradation, mainly in areas with evergreen vegetation. In January and February, the vegetation in the two parks shows different trends in improvement or degradation, with degradation being predominant in People's Park and improvement being predominant in Harmony Park.

In terms of the spatial distribution characteristics of the monthly FVC at each grade in urban parks, apart from the basic stability area, both parks showed that the ratio of the improvement area is the largest from March to April, with the change type being predominantly significant improvement. The spatial distribution characteristics of the March-April' FVC change were significant. The significant improvement areas in both parks were mainly distributed in the middle and outer parts. In People's Park, they corresponded to the south side of the zoo, the peripheral edge of the park, and the inner road trees. In Harmony Park, the significant improvement areas were mainly distributed on the north side of the park, along the water body, and the internal roads. During the periods where degradation was predominant in adjacent months, the ratio of degradation areas was the largest, and the most significant spatial distribution characteristics were as follows: for People's Park, it was from August to September, and for Harmony Park, it was from October to November. In People's Park, the degradation of vegetation in the Harmony Park was mainly slight from August to September,

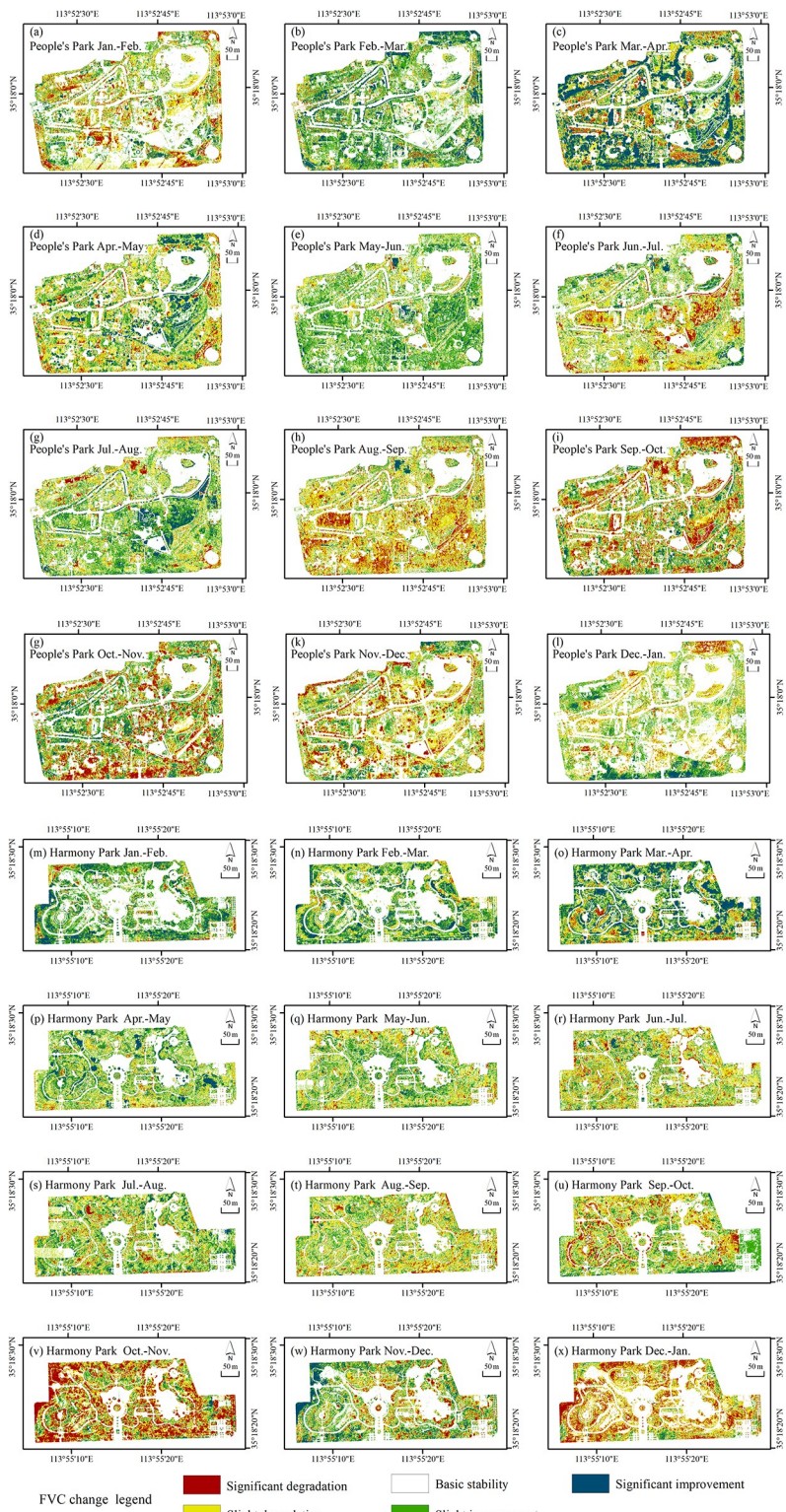

**Fig 8. Dynamic tendency chart of monthly FVC at pixel scale: (a-l) People's Park; (m-x) Harmony Park.**

distributed throughout the park, with the southern side being the most extensive. From October to November, significant degradation was predominant in Harmony Park, mainly distributed around the periphery and central areas, corresponding to the vegetation around the park periphery and water bodies, with the most concentrated degradation occurring on the western periphery. The most significant periods of improvement and degradation in both parks had similar spatial distribution characteristics, mainly concentrated in the peripheral and middle parts of the park. Overall, the dynamic change of monthly FVC at the pixel scale in both parks showed different characteristics. Understanding the monthly FVC change characteristics at pixel scale in urban parks provides a scientific basis for vegetation community management and optimization.

## Discussion

### The impact of park functional distribution on the FVC spatial distribution

The results of this study also demonstrated the close relationship between the FVC spatial distribution characteristics of and the distribution of park functional zones. Overall, the FVC spatial distribution in urban parks showed a pattern of higher values in the periphery and lower values in the center. In conjunction with a ground survey of park vegetation, it was determined that the surrounding vegetation formed a natural barrier, resulting in a complex structure with high plant diversity and enclosed spaces. The interior of the park, designed to accommodate human activities, featured open spaces with amenities such as lawns and plazas, and the vegetation was intentionally kept simple and sparse. These findings aligned with the research findings by Li M, who noted that park areas with heavy human traffic tend to have fewer plants [40]. The design of park vegetation was influenced by functional zones: areas primarily focused on ecological benefits and natural aesthetics tend to have high planting density and abundant greenery, while areas primarily focused on activities and leisure may reduce vegetation density while providing more open space. The study indicated that residents in densely populated urban areas prefer moderately to densely vegetated landscapes due to their interest in wildlife and ecological friendliness [41]. High-density trees increased ground biomass, such as canopies, trunks, and leaves, which in turn obstruct sunlight from reaching the ground, leading to the formation of the cold island effect [42]. Therefore, increasing planting density and subsequently enhancing FVC can lead to higher satisfaction to parks [43, 44].

### The impact of natural disaster disturbances on the FVC change trend

Healthy plants have larger canopies and leaf areas, enabling them to better fulfill ecosystem service functions. However, the health of park vegetation is also threatened by factors such as diseases, pests, improper management, climate change, and extreme weather events [45]. Research on dynamic change in regional-scale monthly FVC mean have also found that the fluctuation in FVC mean over the course of a year showed an overall "∩"-shaped pattern, with FVC values peaking in the months of June to August. Previous studies have also found that regional FVC values reached their maximum in July to September or July to August, and their minimum in January, February, and December [46, 47], which was consistent with the results of this study. In this study, there are differences between the rise and fall of the FVC means and the overall rising and falling trends in both parks from June to August. In July, one park shows a significant increase in bare ground area while the other remains relatively unchanged. Additionally, the peak FVC values for both parks occur at different times. On July 17–20, 2021, Xinxiang experienced several consecutive days of heavy rainfall, with a total rainfall of up to 907mm, leading to severe urban waterlogging. Due to the slow recession of the rainwater, the vegetation in the lower-lying parks remained submerged for an extended period,

resulting in a decrease in FVC values during the submerged period. Furthermore, in this study, water bodies are classified as bare ground, and the fact that the images for both parks were captured on different dates, 26th and 27th of the month, makes the results "seemingly unreasonable". Specifically, in People's Park, the FVC peaks in June and August, while in Harmony Park, the FVC peaks in May and June. People's Park, being lower-lying, ex-perienced extensive flooding of grasslands and shrubs after the water overflowed in July, leading to an increase in bare ground area and a decrease in the FVC mean. Subsequently, in August, as the water receded, the vegetation reappeared, resulting in a decrease in bare ground area and an increase in the average FVC. Harmony Park, being higher in elevation, had relatively unchanged bare ground area in July compared to June due to the recession of the water by the 26th-27th, but the FVC mean was lower due to the stress from the rainfall. By August, the FVC values had mostly recovered to grades similar to the peak in June, but slightly lower than those in May.

### The impact of shooting time on park FVC

The monthly FVC dynamic change analysis at regional scale in urban parks showed that the overall monthly FVC in Harmony Park was greater than in People's Park, with the exception of January. During January-February, People's Park exhibited a trend of degradation, while Harmony Park showed improvement. Upon reviewing the images, it was observed that due to the park's proximity to high-rise buildings, the UAV's shooting time was in the afternoon, resulting in shadows in the January images. Previous studies have indicated that shadows can affect the extraction of vegetation characteristics [48, 49], thus leading to a smaller FVC value in January, contributing to an overall improvement in vegetation from January to February.

### The impact of evergreen vegetation on park FVC

Xinxiang has a warm temperate continental monsoon climate, with deciduous broad-leaved forests as the primary vegetation type. Analysis of the dynamic change in monthly FVC at the pixel scale revealed that the overall improvement or degradation characteristics in the park corresponded to the growth characteristics of deciduous vegetation. Apart from the areas with minimal change, the main types of change aligned with the overall degradation trend. However, there were exceptions in People's Park from April to May and from October to November. From April to May, apart from the areas with minimal change, the largest ratio was in the improvement area, but during this period, the types of FVC degradation and improvement mainly consisted of slight degradation areas. Conversely, from October to November, apart from the areas with minimal change, the predominant trend was degradation, yet the types of FVC degradation and improvement mainly consisted of slight improvement areas. Analysis of the pixel-scale monthly FVC dynamic change maps revealed that the slight degradation areas from April to May and the slight improvement areas from October to November mainly consisted of evergreen vegetation such as ligustrum trees, cypress groves, osmanthus forests, and liriope lawns. It was speculated that the growth period of evergreen vegetation led to the opposite trend between the main change trend and the main change type in periods from April to May and from October to November. Previous studies have also indicated that April is the peak period for evergreen plant withering, and when the ratio of evergreen plants in green spaces is high, the amount of fallen leaves peaks in April [50, 51]. Furthermore, considering that November was the peak growth period for evergreen plants, the high ratio of evergreen plants led to the main change trend and type differing from October to November. The primary vegetation type in Xinxiang is deciduous plants, and evergreen plants, due to their year-round greenery, can compensate for the lack of winter landscapes [52]. Additionally, evergreen

plants typically have a higher annual average FVC, playing an important role in improving winter air quality and the ecological environment [53, 54]. Therefore, it is advisable to appropriately increase the variety and planting area of evergreen plants in parks to enhance FVC.

## Conclusions

1. For the spatial distribution of monthly FVC, both parks exhibit the highest proportion of bare ground area in January and February. People's Park has percentages of 59.17% and 64.46%, while Harmony Park has percentages of 69.10% and 51.92%. The months with the highest proportion of high coverage area are June and August for People's Park (30.71% and 30.26%) and May and June for Harmony Park (34.23% and 33.69%). The spatial distribution of FVC varies each month, but generally, the high and medium-high coverage areas are mainly located at the periphery of the parks, while the medium, medium-low, and low coverage areas are mainly in the central and middle parts of the parks, with bare ground mainly located in the hard ground and water areas. Overall, the spatial distribution of FVC in the parks shows a pattern of high coverage at the periphery and low coverage at the center. This suggests that park development should balance multiple functions and control the proportion of hard ground. Additionally, more tree-lined squares can be created within the parks to cater to both ecological and recreational needs.

2. For the dynamic change of monthly FVC at regional scale, the average FVC changes show an irregular "∩" shape, and the FVC mean values for Harmony Park are generally higher than those for People's Park. The minimum and maximum FVC values for People's Park occur in February (0.17) and August (0.46), while for Harmony Park, they occur in January (0.15) and June (0.50). The areas of bare ground and high coverage areas exhibit the greatest fluctuations in the dynamic changes of FVC area at different grades, with the fluctuation of bare ground and high coverage area showing opposite trends in change and rate of change. The overall vegetation in the parks should enhance the canopy structure of low coverage areas and improve the structure of medium coverage areas to achieve higher FVC and better ecological benefits.

3. For the dynamic change of monthly FVC at pixel scale, the months of February-March, March-April, April-May, May-June, and July-August show improvement as the main trend, while the months of January-February, June-July, August-September, September-October, and October-November show degradation as the main trend. The degradation and improvement are mainly moderate, and overall, the dynamic changes of monthly FVC at pixel scale are relatively stable. The most significant improvement in monthly FVC occurs in March-April for both parks, with the change type being significant improvement. The most significant degradation in FVC occurs in August-September for People's Park and October-November for Harmony Park, with the change type being significant degradation. During the periods of significant improvement and degradation in monthly FVC, significant improvement and degradation are mainly distributed in the pe-riphery and middle parts of the parks. The plant configuration in the parks should strengthen community construction, increase the proportion of evergreen plants, and moderately increase planting density.

## Author Contributions

**Conceptualization:** Yichuan Zhang, Lifang Qiao.

**Data curation:** Yanan Ge.

**Formal analysis:** Yanan Ge.

**Funding acquisition:** Yichuan Zhang, Lifang Qiao.

**Investigation:** Yichuan Zhang, Yanan Ge.

**Methodology:** Yichuan Zhang, Yanan Ge.

**Resources:** Yichuan Zhang, Yanan Ge.

**Software:** Yanan Ge.

**Supervision:** Yichuan Zhang.

**Validation:** Yichuan Zhang.

**Visualization:** Yanan Ge.

**Writing – original draft:** Yanan Ge.

**Writing – review & editing:** Yichuan Zhang, Yanan Ge, Lifang Qiao.

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
