## [Decision Letter · Decision Letter 0]

17 May 2024

PONE-D-24-17066A Study of Spatial Distribution and Dynamic Change in Monthly FVC of Urban ParksPLOS ONE

Dear Dr. Zhang,

Thank you for submitting your manuscript to PLOS ONE. After careful consideration, we feel that it has merit but does not fully meet PLOS ONE’s publication criteria as it currently stands. Therefore, we invite you to submit a revised version of the manuscript that addresses the points raised during the review process.

We look forward to receiving your revised manuscript.

Kind regards,

Pisirai Ndarukwa, Ph.D.

Academic Editor

PLOS ONE

Journal Requirements:

2. In your Methods section, please provide additional information regarding the permits you obtained for the work. Please ensure you have included the full name of the authority that approved the field site access and, if no permits were required, a brief statement explaining why

"This research was funded by the following projects: Key Science and Technology Research and Development Program of Henan Province, China (232102320022), and Key Science and Technology Research and Development Program of Henan Province, China (232102320071)"

5. We note that Figures 1, 3a, 3b, 7a and 7b in your submission contain map images which may be copyrighted. All PLOS content is published under the Creative Commons Attribution License (CC BY 4.0), which means that the manuscript, images, and Supporting Information files will be freely available online, and any third party is permitted to access, download, copy, distribute, and use these materials in any way, even commercially, with proper attribution. For these reasons, we cannot publish previously copyrighted maps or satellite images created using proprietary data, such as Google software (Google Maps, Street View, and Earth). For more information, see our copyright guidelines: http://journals.plos.org/plosone/s/licenses-and-copyright.

    a. You may seek permission from the original copyright holder of Figure(s) [#] to publish the content specifically under the CC BY 4.0 license. 

Please upload the completed Content Permission Form or other proof of granted permissions as an "Other" file with your submission

Reviewers' comments:

Reviewer's Responses to Questions

**Comments to the Author**

1. Is the manuscript technically sound, and do the data support the conclusions?

Reviewer #1: Yes

Reviewer #2: Yes

Reviewer #3: Partly

2. Has the statistical analysis been performed appropriately and rigorously? 

Reviewer #1: Yes

Reviewer #2: Yes

Reviewer #3: Yes

3. Have the authors made all data underlying the findings in their manuscript fully available?

Reviewer #1: Yes

Reviewer #2: Yes

Reviewer #3: Yes

4. Is the manuscript presented in an intelligible fashion and written in standard English?

Reviewer #1: Yes

Reviewer #2: No

Reviewer #3: Yes

5. Review Comments to the Author

Reviewer #1: The manuscript is well-written. Results are systematic and reliable. However, some modifications are needed that are as follows:

• What is the new dimension of the study? Kindly explain explicitly in the objective part.

• The research gaps, methods, results, conclusion should be mentioned in the abstract. It should be like the summary of the study. Some numerical figures may be added.

• The objectives of the study should be clearly written.

• The flowchart of the methodology should be reconstructed. Figure does not meet the required quality.

• Discussion section need more scientific explanation.

• The authors should check the available recent literatures. Some following papers may be discussed for comparative study:


https://doi.org/10.2478/jlecol-2023-0015


https://doi.org/10.1080/23754931.2023.2187314


https://doi.org/10.1080/19479832.2023.2252818


https://doi.org/10.2478/jlecol-2022-0015


http://dx.doi.org/10.1080/24749508.2022.2131962

• Conclusion section needs more clarification.

Reviewer #2: Exploration of the spatial distribution and dynamic change in monthly FVC of region can provides a scientific basis for vegetation management and optimization in this area. This research focuses on two comprehensive parks, using multi-spectral UAV images and the pixel dichotomy model to obtain FVC data monthly and utilizes linear regression analysis to examine the dynamic change of FVC at the park area and pixel scale. The paper was well organized, "Discussion" and "interpretation of results" are basically reasonable. However, the manuscript has to be revised before in published.

1. What is the significance of only comparative analysis of FVC changes in two parks with very small areas and great human interference? What are the theoretical and practical implications of the research results? What are the implications for natural state reserves? This question needs to be well phrased and answered;

2. The techniques and methods in this study, including pixel dichotomy model, linear regression analysis and differencing method, are very mature; Nor do we see any new ideas or perspectives in research. The innovation of this study should be further highlighted.

3. The study pointed out that compared to traditional aerial remote sensing, UAVs have the advantages of flexibility and efficiency, enabling flexible flight planning and control as needed to achieve rapid and precise monitoring of urban green spaces, significantly enhancing research efficiency, however, how to use UAVs to obtain long-term continuous annual scale images of urban green space in larger areas?

4. Please add references to the standard of dividing pure bare area and area completely covered by vegetation in line 105-106.

5. In People’s Park, the highest-coverage area is largest in June and August, while the highest-coverage area is largest in May and June in Harmony Park. Suggest discussing the reasons for the different peak times in the highest-coverage area between the two parks.

6. According to Figure 5 and Figure 6, please analyze and discuss the reasons why the bare area of the two parks in July increased in one case and decreased in the other.

7. The study used the linear regression analysis method and differential method to examine the spatial-temporal dynamic change in monthly FVC, but the result analysis mainly analyzed the temporal change rule and trend of FVC, while the spatial dynamic variation trend analysis is not prominent enough, so it is suggested to emphasize the spatial variation trend analysis.

8. The manuscript requires moderate editing of the English language.

Reviewer #3: The manuscript is quite carefully written and deals with the very important subject of Spatial Distribution and Dynamic Change in Monthly FVC of Urban Parks. This paper employed the pixel dichotomy model to obtain FVC data for each month and utilized linear regression analysis to examine the dynamic change of monthly FVC at the park area scale. Furthermore, the authors employed the differencing method to analyze the dynamic change of monthly FVC at the pixel scale. Overall, the paper is reasonably well organized, and the conclusions are also quite logical. I believe the author has put a lot of effort into this research. However, there are some concerns that need to be improved before it can be published in PLOS ONE. Please find below the detailed comments on improving this paper.

Introduction

Point 1: I recommended that the authors combine the first and second paragraphs of the introduction into one paragraph.

Point 2: In the introduction section, the authors wrote about the importance of urban parks and vegetation cover, but did not explain the innovations of this paper, such as why multispectral images from different months were used, what other scholars have done in the field and their shortcomings. So, the authors should reorganize the introduction section to clarify the innovations of the paper and to add what research scholars at home and abroad have done in the research. The authors can cite the following references to support your points.

https://doi.org/10.1016/j.scitotenv.2022.156990

Materials and Methods

Point 3: I recommend that authors place section 2.2.1 in 2.1 and section 2.2.2 in 2.2 to make the structure of the article clearer.

Point 4: I recommend that the authors number Figure 1 and appropriately enlarge the legend of the figure on the left.

Point 5: In section 2.2.1, the authors should add a table of specific information about the raw imagery, time of acquisition, number of images, flight weather conditions, resolution, etc. In addition, it is recommended that the authors elaborate on the data preprocessing section.

Point 6: I recommend that the authors add a description of the version number of ArcGIS in the 2.2.1 section.

Point 7: The authors mention pixel binary model principle in section 2.2.2 and suggest adding a detailed description of the pixel binary model principal description.

Results

Point 8: The dimensional scale distances in each row of Figure 3 and Figure 7 do not match, and the authors are advised to carefully check the coordinate system for consistent. In addition, I recommend that the authors add clarifications such as (a), (b) and (c) in the corresponding section of the article.

Point 9: Section 3.1.2 is not analyzed in sufficient depth. I recommend that the authors should add a description of this section.

Point 10: The authors mention the temporal variation in Figure 4 in section 3.2.1, but the distribution curve for PP and HP is an approximately normal curve, but the authors use linear regression in the text, please explain the reason for this.

Point 11: The overlapping of the symbols in Figures 5 and 6 leads to unclear presentation and should be adjusted by the authors to ensure better visualization, e.g. by using different colored lines, changing the size of the symbols and adjusting the vertical coordinates. Also, I suggest that the author adjust the color scheme of Figure 6 to make the numbers more visible.

Point 12: I recommended that the authors should add several references to support your comments in the sections 4.2 and 4.3.

https://doi.org/10.1080/15481603.2023.2202506

Discussion

Point 13: The conclusion section is too similar to the abstract section, and I recommend that the authors should revise the conclusion section appropriately.

6. PLOS authors have the option to publish the peer review history of their article (what does this mean?). If published, this will include your full peer review and any attached files.

Reviewer #1: No

Reviewer #2: No

Reviewer #3: No

---

## [Author Response · Author response to Decision Letter 0]

27 May 2024

Dear Editor and Reviewers,

We sincerely appreciate the time and effort you have dedicated to reviewing our manuscript, titled "A Study of Spatial Distribution and Dynamic Change in Monthly FVC of Urban Parks" (Manuscript ID: [PONE-D-24-17066]). We are grateful for your insightful comments and suggestions, which have helped us improve the quality of our work. Below, we have provided detailed responses to each of the reviewers' comments and outlined the changes made in the revised manuscript, and all the changes have been marked in red.

Response to Journal :

Journal Requirements:

Comment 1

Response: We have done preliminary typesetting according to the journal's requirements.

Comment 2

2. In your Methods section, please provide additional information regarding the permits you obtained for the work. Please ensure you have included the full name of the authority that approved the field site access and, if no permits were required, a brief statement explaining why

Response: The parks in this study belong to urban open spaces, and the research aims to provide better service functions for the parks. Therefore, no institutional license is required.

Comment 3

"This research was funded by the following projects: Key Science and Technology Research and Development Program of Henan Province, China (232102320022), and Key Science and Technology Research and Development Program of Henan Province, China (232102320071)"

Response: The funders had no role in study design, data collection and analysis, decision to publish, or preparation of the manuscript

Comment 4

Response: The imagery captured by drones has high resolution and precise geographic coordinates. According to Chinese law, these images can only be used for scientific research and cannot be provided as raw images.

Comment 5

5. We note that Figures 1, 3a, 3b, 7a and 7b in your submission contain map images which may be copyrighted. All PLOS content is published under the Creative Commons Attribution License (CC BY 4.0), which means that the manuscript, images, and Supporting Information files will be freely available online, and any third party is permitted to access, download, copy, distribute, and use these materials in any way, even commercially, with proper attribution. For these reasons, we cannot publish previously copyrighted maps or satellite images created using proprietary data, such as Google software (Google Maps, Street View, and Earth). For more information, see our copyright guidelines: http://journals.plos.org/plosone/s/licenses-and-copyright.

 a. You may seek permission from the original copyright holder of Figure(s) [#] to publish the content specifically under the CC BY 4.0 license. 

Please upload the completed Content Permission Form or other proof of granted permissions as an "Other" file with your submission

Response: The maps we used are standard maps provided by the Map Technology Review Center, Ministry of Natural Resources of China (http://bzdt.ch.mnr.gov.cn/browse.html?picId=%224o28b0625501ad13015501ad2bfc0274%22). According to the website's permission: the public can browse and download standard maps for free, and when using standard maps directly, the surveying and mapping number needs to be marked. Therefore, we added the source and surveying and mapping number in the figure title.

Response to Reviewer 1:

Reviewer #1: The manuscript is well-written. Results are systematic and reliable. However, some modifications are needed that are as follows:

Comment 1

• What is the new dimension of the study? Kindly explain explicitly in the objective part.

Response: We added shortcomings of previous research and the problems to be addressed by this study at the end of the introduction.

Comment 2

• The research gaps, methods, results, conclusion should be mentioned in the abstract. It should be like the summary of the study. Some numerical figures may be added.

Response: We reorganized the abstract into background, methods, results, and conclusions, adding necessary numbers.

Comment 3

• The objectives of the study should be clearly written.

Response: We clearly stated the objectives of the study at the end of the introduction.

Comment 4

• The flowchart of the methodology should be reconstructed. Figure does not meet the required quality.

Response: We created a method flowchart and made adjustments to the figures in the text.

Comment 5

• Discussion section need more scientific explanation.

Response: We have added more explanations and evidence in the discussion section.

Comment 6

• The authors should check the available recent literatures. Some following papers may be discussed for comparative study:


https://doi.org/10.2478/jlecol-2023-0015


https://doi.org/10.1080/23754931.2023.2187314


https://doi.org/10.1080/19479832.2023.2252818


https://doi.org/10.2478/jlecol-2022-0015


http://dx.doi.org/10.1080/24749508.2022.2131962

Response: We consulted some literature and also referred to literature recommended by the reviewers. Based on the content of the research literature and its relevance to the theme of this study, we mainly placed these literature in the introduction section.

Comment 7

• Conclusion section needs more clarification.

Response: We made comprehensive revisions and optimizations to the conclusion section.

Response to Reviewer 2:

Reviewer #2: Exploration of the spatial distribution and dynamic change in monthly FVC of region can provides a scientific basis for vegetation management and optimization in this area. This research focuses on two comprehensive parks, using multi-spectral UAV images and the pixel dichotomy model to obtain FVC data monthly and utilizes linear regression analysis to examine the dynamic change of FVC at the park area and pixel scale. The paper was well organized, "Discussion" and "interpretation of results" are basically reasonable. However, the manuscript has to be revised before in published.

Comment 1

1.What is the significance of only comparative analysis of FVC changes in two parks with very small areas and great human interference? What are the theoretical and practical implications of the research results? What are the implications for natural state reserves? This question needs to be well phrased and answered;

Response: We added relevant explanations at the end of the introduction. We added reasons for selecting these two parks in section 2.1.1. Unlike the natural succession of vegetation in nature reserves, parks are heavily influenced by human activities, with more complex distribution characteristics.

Comment 2

2.The techniques and methods in this study, including pixel dichotomy model, linear regression analysis and differencing method, are very mature; Nor do we see any new ideas or perspectives in research. The innovation of this study should be further highlighted.

Response: Yes, indeed, these techniques and methods are conventional. Our innovation lies in analyzing the differences in FVC (Fractional Vegetation Cover) in different spatial areas at the same time period and the changes in FVC at different time periods in the same spatial area. This way, we can identify the characteristics of "high-quality FVC" and "low-quality FVC" vegetation, providing reference for park vegetation planning.

Comment 3

3.The study pointed out that compared to traditional aerial remote sensing, UAVs have the advantages of flexibility and efficiency, enabling flexible flight planning and control as needed to achieve rapid and precise monitoring of urban green spaces, significantly enhancing research efficiency, however, how to use UAVs to obtain long-term continuous annual scale images of urban green space in larger areas?

Response: Drone technology has seen rapid development in China in recent years and has been widely used in agricultural and forestry monitoring. The parks we studied are not large in scale, and data acquisition only takes half an hour per month, so the resolution is indeed high. If larger-scale, long-term continuous monitoring is needed, flying at higher altitudes with long-endurance batteries can still achieve high-resolution imagery within an acceptable time frame.

Comment 4

4.Please add references to the standard of dividing pure bare area and area completely covered by vegetation in line 105-106.

Response: We added explanations about the classification criteria in section 2.2.2.

Comment 5

5.In People’s Park, the highest-coverage area is largest in June and August, while the highest-coverage area is largest in May and June in Harmony Park. Suggest discussing the reasons for the different peak times in the highest-coverage area between the two parks.

Response: We explained the reasons for the inconsistency in peak times in section 4.2, mainly influenced by heavy rainfall and the time of data collection.

Comment 6

6.According to Figure 5 and Figure 6, please analyze and discuss the reasons why the bare area of the two parks in July increased in one case and decreased in the other.

Response: We explained in section 4.2. The area of bare land in People's Park did increase, while in Harmony Park it remained basically unchanged, with no significant decrease.

Comment 7

7.The study used the linear regression analysis method and differential method to examine the spatial-temporal dynamic change in monthly FVC, but the result analysis mainly analyzed the temporal change rule and trend of FVC, while the spatial dynamic variation trend analysis is not prominent enough, so it is suggested to emphasize the spatial variation trend analysis.

Response: We added relevant explanations in sections 3.2.2 and 3.3.2.

Comment 8

8. The manuscript requires moderate editing of the English language.

Response: We polished the article in English.

Response to Reviewer 3:

Reviewer #3: The manuscript is quite carefully written and deals with the very important subject of Spatial Distribution and Dynamic Change in Monthly FVC of Urban Parks. This paper employed the pixel dichotomy model to obtain FVC data for each month and utilized linear regression analysis to examine the dynamic change of monthly FVC at the park area scale. Furthermore, the authors employed the differencing method to analyze the dynamic change of monthly FVC at the pixel scale. Overall, the paper is reasonably well organized, and the conclusions are also quite logical. I believe the author has put a lot of effort into this research. However, there are some concerns that need to be improved before it can be published in PLOS ONE. Please find below the detailed comments on improving this paper.

Introduction

Comment 1

Point 1: I recommended that the authors combine the first and second paragraphs of the introduction into one paragraph.

Response: We merged relevant content.

Point 2: In the introduction section, the authors wrote about the importance of urban parks and vegetation cover, but did not explain the innovations of this paper, such as why multispectral images from different months were used, what other scholars have done in the field and their shortcomings. So, the authors should reorganize the introduction section to clarify the innovations of the paper and to add what research scholars at home and abroad have done in the research. The authors can cite the following references to support your points.

https://doi.org/10.1016/j.scitotenv.2022.156990

Response: We summarized the characteristics of previous research at the end of the introduction and clarified the innovation of the study.

Materials and Methods

Point 3: I recommend that authors place section 2.2.1 in 2.1 and section 2.2.2 in 2.2 to make the structure of the article clearer.

Response: We adjusted relevant content.

Point 4: I recommend that the authors number Figure 1 and appropriately enlarge the legend of the figure on the left.

Response: We numbered the figures and labeled the sources of the drawings.

Point 5: In section 2.2.1, the authors should add a table of specific information about the raw imagery, time of acquisition, number

---

## [Decision Letter · Decision Letter 1]

31 Jul 2024

A Study of Spatial Distribution and Dynamic Change in Monthly FVC of Urban Parks

PONE-D-24-17066R1

Dear Dr. Zhang,

We’re pleased to inform you that your manuscript has been judged scientifically suitable for publication and will be formally accepted for publication once it meets all outstanding technical requirements.

Kind regards,

Pisirai Ndarukwa, Ph.D.

Academic Editor

PLOS ONE

Additional Editor Comments (optional):

Reviewers' comments:

Reviewer's Responses to Questions

**Comments to the Author**

1. If the authors have adequately addressed your comments raised in a previous round of review and you feel that this manuscript is now acceptable for publication, you may indicate that here to bypass the “Comments to the Author” section, enter your conflict of interest statement in the “Confidential to Editor” section, and submit your "Accept" recommendation.

Reviewer #1: All comments have been addressed

Reviewer #3: (No Response)

2. Is the manuscript technically sound, and do the data support the conclusions?

Reviewer #1: Yes

Reviewer #3: (No Response)

3. Has the statistical analysis been performed appropriately and rigorously? 

Reviewer #1: Yes

Reviewer #3: (No Response)

4. Have the authors made all data underlying the findings in their manuscript fully available?

Reviewer #1: Yes

Reviewer #3: (No Response)

5. Is the manuscript presented in an intelligible fashion and written in standard English?

Reviewer #1: Yes

Reviewer #3: (No Response)

6. Review Comments to the Author

Reviewer #1: The revised manuscript is much improved than the original manuscript. The authors have successfully justified the reviewers' comments and incorporated the required corrected portions in the revised manuscript. The methods are cklear. The results are relevant and logical. The reference list is updated. The image quality has been improved a lot. Now, it can be recommended for publication.

Reviewer #3: (No Response)

7. PLOS authors have the option to publish the peer review history of their article (what does this mean?). If published, this will include your full peer review and any attached files.

Reviewer #1: No

Reviewer #3: No

---

## [Editor Report · Acceptance letter]

14 Aug 2024

PONE-D-24-17066R1 

PLOS ONE

Dear Dr. Zhang, 

I'm pleased to inform you that your manuscript has been deemed suitable for publication in PLOS ONE. Congratulations! Your manuscript is now being handed over to our production team.

Kind regards, 

on behalf of

Prof Pisirai Ndarukwa 

Academic Editor

PLOS ONE